# Intraoperative Assessment of Surgical Stress Response Using Nociception Monitor under General Anesthesia and Postoperative Complications: A Narrative Review

**DOI:** 10.3390/jcm11206080

**Published:** 2022-10-14

**Authors:** Munetaka Hirose, Hiroai Okutani, Kazuma Hashimoto, Ryusuke Ueki, Noriko Shimode, Nobutaka Kariya, Yumiko Takao, Tsuneo Tatara

**Affiliations:** 1Department of Anesthesiology and Pain Medicine, Hyogo Medical University School of Medicine, Nishinomiya 663-8501, Japan; 2Pain Clinic, Hyogo Medical University Hospital, Nishinomiya 663-8501, Japan

**Keywords:** anesthetic, postoperative morbidity, surgical procedure

## Abstract

We present a narrative review focusing on the new role of nociception monitor in intraoperative anesthetic management. Higher invasiveness of surgery elicits a higher degree of surgical stress responses including neuroendocrine-metabolic and inflammatory-immune responses, which are associated with the occurrence of major postoperative complications. Conversely, anesthetic management mitigates these responses. Furthermore, improper attenuation of nociceptive input and related autonomic effects may induce increased stress response that may adversely influence outcome even in minimally invasive surgeries. The original role of nociception monitor, which is to assess a balance between nociception caused by surgical trauma and anti-nociception due to anesthesia, may allow an assessment of surgical stress response. The goal of this review is to inform healthcare professionals providing anesthetic management that nociception monitors may provide intraoperative data associated with surgical stress responses, and to inspire new research into the effects of nociception monitor-guided anesthesia on postoperative complications.

## 1. Introduction

Patients sometimes have discomforts (e.g., pain, nausea and vomiting) and complications after surgery. Especially higher invasiveness of surgery induces a higher degree of stress responses, which are associated with postoperative complications [1,2,3]. Since postoperative complications worsen patient outcomes and burden hospital finances [4,5,6], several tools have been developed to predict the incidence of postoperative complications and mortality before and after surgery.

The Physiological and Operative Severity Score for the Enumeration of Mortality and Morbidity (POSSUM) system is scored using preoperative physiological variables and operative severity variables, including operative magnitude, number of operations within the preceding 30 days, presence of malignancy, and the elective/emergency nature of interventions, in addition to intraoperative variables of blood loss and peritoneal contamination [7]. On the other hand, the American College of Surgeons National Surgical Quality Improvement Program (ACS NSQIP) surgical risk calculator uses preoperative variables and surgical procedure codes before surgery to predict the incidence of major postoperative complications [8]. In addition, the Surgical Mortality Probability Model (S-MPM) [9] and the Surgical Outcome Risk Tool (SORT) [10] predict the incidence of postoperative mortality using a severity score for the surgical procedure with preoperative variables. To date, severity of surgery has been scored preoperatively according to the name of the surgical procedure in existing prediction models. Furthermore, objective intraoperative data are required for better risk prediction of postoperative complications at the end of surgery [11,12,13]. Intraoperative quantification of surgical severity would thus be valuable for predicting postoperative complications more accurately and for better intraoperative management to suppress postoperative complications. A method of quantitatively assessing surgical invasiveness has also been under development [14].

This narrative review proposes the use of nociception monitors, which assess a balance between nociception caused by invasiveness of surgery and anti-nociception provided by anesthesia in patients under general anesthesia, to provide intraoperative quantitative values for assessing surgical stress responses. One may think, however, that both surgical severity and inflammatory response that may adversely affect postoperative outcome is not always related to nociception, and also that nociception is not the major source of surgical stress response and systemic inflammation that may cause detrimental outcome. By filling the gaps between nociception, inflammation, and consequently postoperative complications, this review provides a new concept that nociception monitor may allow a glimpse of the surgery related stress response apart from its original role.

## 2. Surgical Trauma, Nociception and Anti-Nociception

Surgical trauma evokes noxious stimuli of mechanical stimulation, pressure, and inflammatory mediators, activating nociceptors on peripheral somatosensory neurons and thus inducing nociception, which is the neural process of encoding noxious stimuli [2,15].

On the other hand, anti-nociception is one of the roles of general anesthesia during surgery, in addition to unconsciousness, amnesia, and akinesia. Since pain represents conscious perception of nociceptive information, pain is not obvious during surgery under general anesthesia [16]. Nociception caused by intense noxious stimuli in the skin, however, can reach the cerebral cortex even under deep general anesthesia using propofol with or without remifentanil [17,18], and thus is still evoked even during general anesthesia.

Nociceptive information ascends from myelinated Aδ and unmyelinated C fibers through the spinothalamic tract to the thalamus and cerebral cortex [19]. Collateral branches below the bulbar-pontine junction send noxious information to the vasomotor center in the rostal ventrolateral medulla, activating the somato-sympathetic reflex and causing increases in blood pressure and heart rate. Such sympathetic activities during surgery under general anesthesia receive feedback regulations from the descending pain inhibitory pathway and baroreflex function [19,20,21,22] and are also suppressed by anesthetic management (Figure 1).

## 3. Nociception Monitors

When the state of general anesthesia is inadequate for the level of nociception during surgery and corresponds to inadequate anti-nociception, heart rate and blood pressure increase, alerting the anesthesiologist to the possibility of increased nociception [23]. Changes in blood pressure or heart rate, however, can be caused not only by nociception, but also by posture, bleeding, cardiovascular agents, respiratory state, and body temperature. The specificity of hemodynamic changes as a nociception monitor has thus proven inadequate [24,25].

In addition to blood pressure and heart rate, other indirect signs of nociception (e.g., heart rate variability, plethysmogram amplitude, pupillary reflex dilatation, respiratory rate, and body movement) have been used to assess nociception under general anesthesia [26]. Each of these signs, as physiological or pathophysiological responses to nociception, change according to the greater or lesser extent of noxious stimuli. Neurophysiological responses to noxious stimuli, such as changes in electroencephalography (EEG) activity and EEG response entropy, have therefore also been investigated in patients under general anesthesia [27,28,29,30]. Noxious stimuli under general anesthesia are associated with increased delta power and decreased alpha power in frontal leads from a power spectrum analysis of EEG [28,30]. The clinical significance of neurophysiological responses including EEG responses, however, remains unclear [30,31]. So far, direct methods to monitor nociception clinically in patients under general anesthesia do not exist.

Although nociception by itself is difficult to measure clinically in unconscious patients [32], combining information from different sources was found to allow development of a nociception monitor, which promises more accurate reflection of nociception than traditionally used indirect signs of nociception [33]. Huiku et al. elaborated the surgical pleth index (SPI), originally termed the surgical stress index, corresponding to nociceptive stimuli under general anesthesia [34,35,36]. The SPI is calculated using two variables: heartbeat intervals; and plethysmographic pulse wave amplitude. This represents the balance between nociception due to surgical trauma and anti-nociception provided by anesthesia. Further, several nociception monitors have been developed, including the analgesia nociception index (ANI) using heart rate variability; the nociception level (NoL) using photoplethysmography, galvanic skin response, temperature, and an accelerometer; the nociceptive response (NR) using heart rate, systolic blood pressure, and perfusion index; and the pupillary pain index (PPI) using pupillometry (Table 1) [32,37,38]. These nociception monitors have been anticipated to allow assessment of the balance between nociception and anti-nociception under general anesthesia (Figure 1) [32,38,39].

## 4. Disadvantages of Nociception Monitors for Assessing the Balance between Nociception and Anti-Nociception

Although the sensitivity of some of these nociception monitors in response to nociceptive stimuli is likely higher than traditional vital signs, such as blood pressure or heart rate, they have several disadvantages as quantitative monitors of balance between nociception and anti-nociception. Either cardiovascular agents or hemodynamic changes affect values in nociception monitors when autonomic responses are used as sources for monitoring, and either sedatives, muscle relaxants, or neurological comorbidity also affect values in nociception monitors when EEG and electromyography are utilized in monitoring. As a result, specificity in terms of not reflecting responses to non-nociceptive stimuli in these nociception monitors is unfortunately still no better than that of traditional physiological responses [30,32,39].

Currently, no standard measurements have been defined for maximum or minimum nociception under general anesthesia. That is, values from nociception monitors, which range between 0 and 1, 10 or 100, are settled expediently and empirically. For example, stimulus intensities were defined to develop the NoL, as: no noxious stimulus = 0; minor noxious stimulus = 1–2; moderate noxious stimulus (small skin incision) = 3–4; severe noxious stimulus (large skin incision) = 5–6; and extreme noxious stimulus = 7–10 [47].

Finally, in clinical investigations that examined the effects of nociception monitor-guided anesthesia on opioid consumption and stress hormone levels during surgery and also on postoperative pain, results differed between nociception monitors [48,49]. In patients undergoing radical retropubic prostatectomy, both NoL-guided and PPI-guided opioid administrations reduced intraoperative remifentanil consumption with elevated serum levels of cortisol after surgery, although both SPI-guided and standard opioid administrations showed increased intraoperative remifentanil consumption with decreased serum levels of cortisol after surgery. Levels of postoperative pain, however, were comparable between these four groups [49]. Other studies examining the effects of nociception monitor-guided anesthesia on opioid consumption during surgery also showed different results among nociception monitors [48,50,51]. These differences suggest that each nociception monitor using different variables would assess different aspects of neurophysiological or physiological responses to noxious stimuli in addition to the balance between nociception and anti-nociception under general anesthesia.

Monitoring nociception during surgery under general anesthesia has been proposed to avoid or manage sudden hemodynamic changes or unexpected body movements by monitoring the balance between nociception and anti-nociception. Although these monitors are more sensitive in detecting autonomic changes in nociceptive stimuli than traditionally used parameters, such as BP or HR, there is currently no convincing clinical evidence in favor of these monitors in that regard. Since these nociception monitors reflect autonomic responses elicited by both the balance between nociception and anti-nociception and the balance between inflammation and anti-inflammation, such monitoring is anticipated to have further roles to play in optimizing patient outcomes by monitoring other aspects of nociception (e.g., surgical stress responses) (Figure 1).

## 5. Advantages of Nociception Monitors for Assessing Surgical Stress Responses

Surgical trauma stimulates both the nociceptors of the somatosensory nervous system and immune cells, eliciting surgical stress responses that include activation of the hypothalamic–pituitary–adrenal axis, sympathetic nervous system, and immune responses, to maintain physiological homeostasis perioperatively [1,2,3,52,53]. The degree of these integrated responses is determined by the invasiveness, and duration of surgery. Surgical stress response represents a physiological and pathophysiological response to surgical trauma, comprising both neuroendocrine-metabolic and inflammatory-immune responses (Figure 1) [3].

The neuroendocrine-metabolic response involves the sympathetic nervous system, endocrine system, and metabolic responses [3,46]. Sympathetic nervous system activation induces hemodynamic changes caused by increases in systemic vascular resistance, arterial blood pressure, and heart rate, and also induces hyperglycemia. Endocrine system response activates the hypothalamic–pituitary–adrenal axis, which increases plasma concentration of cortisol, in addition to increased secretions of growth hormone, antidiuretic hormone, and thyroxine. These hemodynamic and endocrine responses to surgical trauma coordinate with each other through a feedforward or feedback loop to maintain plasma volume and cardiovascular homeostasis, and also to meet the elevated oxygen demand arising during surgery. Metabolic responses to surgical trauma include both metabolic and catabolic processes, which sustain energy production and provide substrates for the healing process.

Ledowski and Chen et al. examined associations between nociception monitor values, stress hormone levels, and autonomic nervous system activity, and showed that SPI values during surgery under general anesthesia correlated with blood concentrations of cortisol, norepinephrine, and epinephrine [40,41]. SPI values, which are calculated using heartbeat intervals and plethysmographic pulse wave amplitude, also reportedly correlated with autonomic nervous system activities [42]. Furthermore, ANI and PPI monitors, of which sources are heart rate variability and pupillometry, respectively, have also been thought to depend on autonomic nervous system activities (Table 1) [40,41,42]. The intraoperative role for nociception monitor in assessing surgical stress responses, however, has been undervalued so far.

On the other hand, the inflammatory-immune response, which is induced by the balance between inflammation caused by surgical trauma and anti-inflammation due to anesthetic management using non-steroidal anti-inflammatory drug (NSAID) or steroid (Figure 1), produces pro-inflammatory cytokines (e.g., interleukin (IL)-6, IL-1β, IL-8, tumor necrosis factor-α), anti-inflammatory cytokines (e.g., IL-4, IL-10, transforming factor-β), and acute-phase proteins (e.g., C-reactive protein (CRP), fibrinogen, D-dimer) [3,54]. Within these proteins, peak blood concentrations of IL-6 and CRP after surgery are associated with the invasiveness of surgical stress [55]. In a previous study, averaged NR values during surgery were reportedly associated with CRP levels after gastrointestinal surgery (Table 1) [43]. In addition, associations have been identified between inflammatory and autonomic responses [56,57,58]. Increases in blood concentrations of CRP, IL-6, and fibrinogen reportedly correlate with increases in resting heart rate [59].

To summarize the above descriptions, reciprocal regulations exist among the balance between nociception and anti-nociception, the balance between inflammation and anti-inflammation, neuroendocrine-metabolic response, and inflammatory-immune response during surgery under general anesthesia. Under anesthetic management, using short-acting opioid, regional anesthesia, and β-adrenergic blocker suppresses neuroendocrine-metabolic responses, and both NSAID and β-adrenergic blocker counteract inflammatory-immune response [2,16]. Despite being originally developed to assess the balance between nociception and anti-nociception, nociception monitors (especially those using autonomic responses) reflect physiological and pathophysiological responses to surgical stimuli, and so could be used to assess additional aspects of surgical stress responses under general anesthesia (Figure 1).

## 6. Postoperative Complications and Surgical Stress Responses

Postoperative complications exert negative influence on patient outcomes, and burden hospital resources in perioperative care [4,5,6]. The occurrence of postoperative complications, including cardiopulmonary complications, infections, cerebrovascular complications, and renal dysfunction, impact 30-day mortality rates and long-term survival after various surgeries [59,60,61,62]. Perioperative management to suppress postoperative complications is required for the optimization of both the physical condition of patients after surgery and health-care costs. Although the underlying mechanisms by which postoperative complications aggravate short- and long-term survival remain uncertain, physiological and pathophysiological responses to surgical trauma might be among the candidate mechanisms.

Surgical stress responses are evoked to facilitate homeostasis and support postoperative recovery. On the other hand, stress responses are exacerbated by an excessive degree of surgical trauma and preoperative morbidity, then the inadequate surgical stress response potentially causes cognitive and cardiac dysfunction, vascular instability, endothelial activation, inflammation, coagulopathy, and immunosuppression, resulting in major postoperative complications and death (Figure 1) [2]. The degree of surgical trauma and preoperative morbidity also influence inflammatory and immune responses to surgery, where local inflammation may lead to systemic inflammatory responses with pro-inflammatory and immunosuppressive states [54]. These impaired immune responses to surgical trauma are associated with poor outcomes [3,63].

Pro-inflammatory cytokines also disrupt the integrity of the blood–brain barrier and induce neuroinflammation, impairing cognitive function perioperatively. Particularly in elderly patients after cardiac or orthopedic surgery, the incidences of postoperative delirium and postoperative cognitive dysfunction are relatively high, and these cognitive dysfunctions are associated with an increased incidence of postoperative complications [63,64].

Several methods of perioperative management have been utilized to support postoperative recovery by reducing surgical stress responses. Enhanced recovery after surgery (ERAS) programs have been developed to accelerate patient recovery after major surgery, by utilizing pre-, intra-, and postoperative management strategies to mitigate surgical stress responses [65]. Intraoperative strategies to reduce stress responses include reduction in nociception using opioids and regional anesthesia, minimally invasive surgery, pharmacological reduction in inflammatory response, and prevention of heat loss [3,52,65,66]. Implementation of the ERAS program reduces postoperative CRP levels [67] and improves postoperative outcomes, while also reducing hospital costs in patients undergoing various surgeries of the abdomen [68,69,70] and spine [71]. In addition, prehabilitation, representing a multimodal program combining exercise, nutrition and psychological interventions for patients before surgery, is also expected to improve surgical outcomes by optimizing the ability to withstand surgical stress [53,72]. The adequacy of ERAS and/or prehabilitation programs may be monitored through nociception monitors, which need to be validated.

## 7. Nociception Monitors and Postoperative Complications

Even single hemodynamic parameters during surgery, such as higher or lower blood pressure, higher heart rate, or lower peripheral perfusion index, are reportedly associated with major postoperative complications after non-cardiac surgery, including acute kidney injury, delirium, myocardial injury, stroke, and sepsis [73,74,75,76,77], and with those defined as Clavien–Dindo class ≥ IIIa [78]. The Clavien–Dindo classification for grading postoperative complications within 30-days of surgery includes seven grades: grade I, any deviation from the normal postoperative course; grade II, normal course altered; grade IIIa, complications that require interventions performed under local anesthesia; grade IIIb, complications that require interventions performed under general or epidural anesthesia; grade IVa, life-threatening complications with single organ dysfunction; grade IVb, life-threatening complications with multi-organ dysfunction; and grade V, death [79].

The combination of these intraoperative hemodynamic parameters has also been proposed to predict postoperative complications. A lower score on the 10-point Surgical Apgar Score, in which the lowest mean blood pressure during surgery is lower than a standard value and the lowest heart rate during surgery is higher than a standard value, predicts major postoperative complications after colon surgery [80,81] and after non-cardiac surgery [82]. In patients undergoing orthotopic liver transplantation, logistic regression using intraoperative hemodynamic data with preoperative comorbidity can reportedly predict major complications and mortality [83]. Results from nociception monitors combining these autonomic responses are thus also expected to be associated with postoperative complications. Higher values of NR averaged from the start to the end of surgery (mean NR), which are calculated using heart rate, systolic blood pressure, and peripheral perfusion index, were reportedly associated with a higher incidence of major postoperative complications, defined as a Clavien–Dindo Class ≥ IIIa, after gastrointestinal surgery (Table 1) [44]. Mean NR values also increased in the order of severity of postoperative complications in patients undergoing total shoulder arthroplasty, gastrointestinal surgery, and thoracic surgery (Figure 2) [44,45,46,84,85].

Further studies are required to examine whether other nociception monitor values could provide an intraoperative objective index for monitoring surgical stress responses during surgery, which correlates with postoperative complications.

## 8. Nociception Monitor-Guided Anesthesia for Suppressing Surgical Stress Responses

In patients undergoing radical retropubic prostatectomy, SPI-guided anesthesia reportedly attenuated intraoperative increases in adrenocorticotropic hormone (ACTH) and cortisol levels more than standard anesthesia without using a nociception monitor [45]. Conversely, NoL-guided anesthesia, where NoL values are based on photoplethysmography, galvanic skin response, temperature, and an accelerometer, augmented these surgical stress responses in patients undergoing the same surgery [49]. One reason for these differences in surgical stress responses during nociception monitor-guided anesthesia might be that only intraoperative opioid doses were adjusted within the target range of each nociception monitor value. Since intraoperative multimodal approaches, including use of regional anesthesia with avoidance of long-acting opioids, intraoperative normothermia, maintaining fluid balance, and restrictive use of drains and tubes, have been utilized to reduce postoperative complications by suppressing surgical stress responses [86,87], the adequacy of nociception monitor-guided multimodal general anesthesia may be relevant for suppressing surgical stress responses, which need to be validated.

## 9. Conclusions

Both higher levels of nociception due to surgical invasiveness and preoperative worse morbidity exacerbate surgical stress responses, and inversely both perioperative management and anti-nociception provided by anesthesia suppress them. Since inadequate surgical stress responses occasionally cause major postoperative complications, intraoperative objective monitors for the assessment of surgical stress responses have been anticipated for predicting postoperative complications, and also for better intraoperative management of anesthesia. Nociception monitors, particularly monitoring autonomic responses during surgery under general anesthesia, were originally developed to assess the balance between nociception and anti-nociception but have potential to monitor surgical stress responses including neuroendocrine-metabolic and inflammatory-immune responses. Although the monitors more accurately reflect nociception than the traditionally used hemodynamic parameters such as blood pressure or heart rate, there is no solid evidence regarding any clinically relevant influence of these devices on patient outcome. Since few studies have reported associations between nociception monitor values and postoperative complications, further investigations are warranted to clarify associations between nociceptive monitor values, surgical stress responses, and postoperative complications, with a view to optimizing nociception monitor-guided multimodal general anesthesia for the suppression of surgical stress responses.

## Figures and Tables

**Figure 1 jcm-11-06080-f001:**
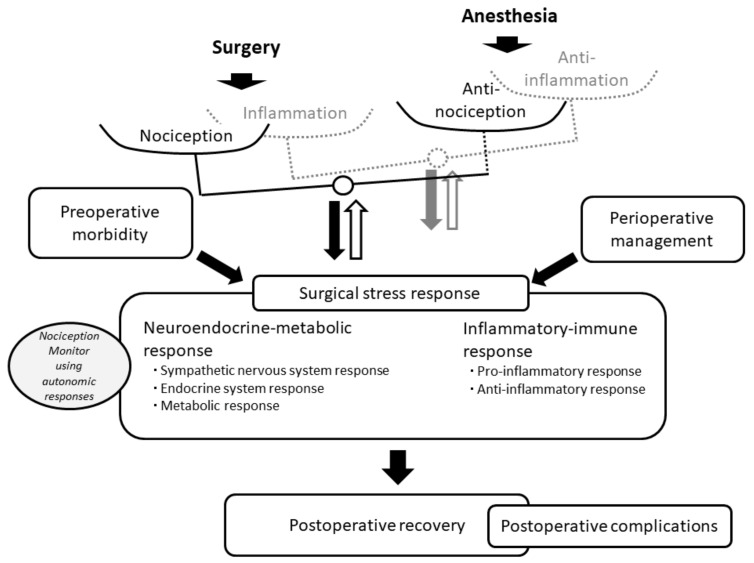
Values assessed by nociception monitors during surgery under general anesthesia. The degree of balance between nociception caused by surgical trauma and anti-nociception provided by anesthesia affects surgical stress responses, consisting of both neuroendocrine-metabolic and inflammatory-immune responses. The degree of balance between inflammation caused by surgical trauma and anti-inflammation due to anesthetic management, preoperative morbidity, and perioperative management (e.g., ERAS, prehabilitation) also affect surgical stress responses. On the other hand, the balance between nociception and anti-nociception receives feedback regulation from the descending pain inhibitory pathway and baroreflex function (white arrow with black outline). The balance between inflammation and anti-inflammation also receives feedback regulation from the endocrine response (e.g., cortisol) (white arrow with gray outline). The surgical stress response plays a role in sustaining patient homeostasis and supporting postoperative recovery. An excessive degree of surgical trauma and preoperative morbidity, however, exacerbates surgical stress responses, potentially causing postoperative complications. Nociception monitors using autonomic responses (e.g., ANI, SPI, NoL, NR, PPI) represent different aspects of intraoperative surgical stress, in addition to the balance between nociception and anti-nociception. ANI; analgesia nociception index, ERAS; enhanced recovery after surgery, NoL; nociception level, NR; nociceptive response, PPI; pupillary pain index, SPI; surgical pleth index.

**Figure 2 jcm-11-06080-f002:**
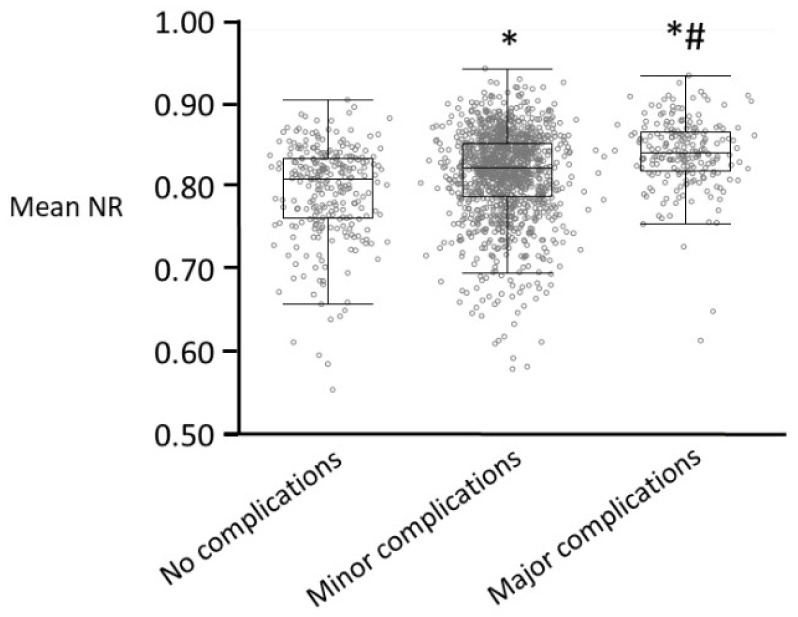
Box-and-whisker plots of averaged values of NR from the start to the end of surgery (mean NR) in patients with no complications (n = 263), minor complications (Clavien–Dindo classification = I or II, n = 1201), and major complications (Clavien–Dindo classification ≥ IIIa, n = 212) after surgery. Gray circles show individual data points. Data in this graph were obtained from [44,45,46,84,85]. Significant differences are defined at * *p* < 0.001 vs. no complications, and ^#^ *p* < 0.001 vs. minor complications. NR; nociceptive response.

**Table 1 jcm-11-06080-t001:** Nociception monitors using autonomic responses.

Nociception Monitor	Sources of Measurement [32,37,38]	Surgical Stress Responses Correlating with Nociception Monitor Values [40,41,42,43]	Surgical Procedures for which Incidence of Postoperative Complications Correlates with Nociception Monitor Values [44,45,46]
ANI	Heart rate variability	Parasympathetic activity	-
NoL	Accelerometry Galvanic skin response Photoplethysmography Temperature	-	-
NR	Heart rate Perfusion index Systolic blood pressure	CRP	Gastrointestinal surgery Lung resection surgery
PPI	Pupillometry	Sympathetic activity	-
SPI	Heartbeat intervals Plethysmographic amplitude	Cortisol Epinephrine Norepinephrine Sympathetic activity	-

ANI; analgesia nociception index, NoL; nociception level, NR; nociceptive response, PPI; pupillary pain index, SPI; surgical pleth index.

## Data Availability

The data that support the findings of this study are available on request from the corresponding author.

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
