# Peer review of "Intraoperative Assessment of Surgical Stress Response Using Nociception Monitor under General Anesthesia and Postoperative Complications: A Narrative Review"

_jcm, 2022, doi:10.3390/jcm11206080_

Round 1
Reviewer 1 Report
The authors submitted a narrative review regarding the potential role of nociception monitors to assess intraoperative stress response that may be related to complications. Below are my comments that the authors may consider in revising the manuscript.
Abstract
14: and occasionally induces à which are associated with the occurrence of
15: suppresses à mitigates (or attenuates)
16: Add sentence before ‘The original role~‘: Furthermore, improper attenuation of nociceptive input and related autonomic effects may induce increased stress response that may adversely influence outcome even in minimally invasive surgeries.
17: would thus correspond to à may allow
18-20: We propose ~ complications. delete sentence.
21: provide à may provide
Introduction
29: causing à which are associated with
44-46: delete sentence
50-51: To date, severity of surgery ~prediction models. Move this sentence to line 44
46-56: Fundamental problem is that surgical severity and inflammatory response that may adversely effect outcome is not always related to nociceptive input. Also, nociceptive input is not the major source of stress response and systemic inflammation that may cause detrimental outcome. Likewise, the link between nociception and inflammation and consequently complications is unclear. Therefore, the whole paragraph should be rephrased to indicate that despite these uncertainties, nociception monitors may allow a glimpse of the surgery related stress response apart from its original purpose of monitoring the balance between nociception and proper anesthetic management.
Nociception monitors
109: qNOX is unique in that it does not rely on a measure of autonomic activity. Therefore, the remark regarding the Table 1 can be misleading as it describes nociception monitors using autonomic responses. Clear distinctions and descriptions should be provided regarding the monitors that rely on autonomic responses or not.
Disadvantages of nociception monitors
Before 165 (Such monitoring~), add a sentence that clearly states the following contents: Although these monitors are more sensitive in detecting autonomic changes in nociceptive stimuli than traditionally used parameters, such as BP or HR, there is currently no convincing clinical evidence in favor of these monitors in that regard.
167: it should be noted that the potential to detect surgical stress response would only be confined to nociception monitors detecting autonomic responses, theoretically.
Advantages of nociception monitors
193: Although it is displayed in the Table, a brief description regarding the assessed variables of the SPI should be added.
194: Likewise, descriptions regarding the assessed variables of ANI and PPI should be added.
Postoperative complications and surgical stress response
220: reduce à exert negative influence on
246: enhanced à Enhanced
257-260: there is no clear link between ERAS (including prehabilitation) and the utility of intraoperative nociception monitors in that regard. The sentence needs to be rephrased that the adequacy of ERAS may be monitored through nociception monitors, which need to be validated.
Nociception monitors and postoperative complications
288-295: average NR values showed correlation with the order of the surgical risks, which was already known (and classified) as low-, intermediate, and high-risk before surgery. What additional clinical value does the nociception monitor provide in that regard? Please, comment.
Nociception monitor-guided anesthesia
308: conversely was used twice. Delete the second one.
308: brief description of the assessed variables by the NoL should be included
317: it should be clearly noted that this notion also needs to be validated
Conclusions
326: hemodynamic data à how about monitoring autonomic responses in general?
330: before the start of the sentence ‘Since few studies~’, add a sentence that clearly describes the pros and cons of the monitors: Although the monitors more accurately reflect nociception than the traditionally used hemodynamic parameters such as blood pressure or heart rate, there is no solid evidence regarding any clinically relevant influence of these devices on patient outcome.
Author Response
Response to Reviewer 1 Comments
Following to the reviewers’ comments, we rewrote the whole manuscript. Our revised marked copy of manuscript highlighted the changes in red color.
The Review Report (Reviewer 1)
Point 1: 14: and occasionally induces à which are associated with the occurrence of
Response 1: Thank you very much for your valuable comments. We changed “occasionally induces major postoperative complications” to “which are associated with the occurrence of major postoperative complications” in the revised manuscript.
Point 2: 15: suppresses à mitigates (or attenuates)
Response 2: We changed “suppresses” to “mitigates” in the revised manuscript.
Point 3: 16: Add sentence before ‘The original role~‘: Furthermore, improper attenuation of nociceptive input and related autonomic effects may induce increased stress response that may adversely influence outcome even in minimally invasive surgeries.
Response 3: We added the above sentence in the revised manuscript.
Point 4: 17: would thus correspond to à may allow
Response 4: We changed “would thus correspond to” to “may allow” in the revised manuscript.
Point 5: 18-20: We propose ~ complications. delete sentence.
Response 5: We deleted this sentence.
Point 6: 21: provide à may provide
Response 6: We changed “provide” to “may provide”.
Point 7: 29: causing à which are associated with
Response 7: We changed “causing” to “which are associated with” in the revised manuscript.
Point 8: 44-46: delete sentence
Response 8: We deleted this sentence with two references of #11and #12.
Point 9: 50-51: To date, severity of surgery ~prediction models. Move this sentence to line 44
Response 9: We moved this sentence to lines 43-45 in the revised manuscript.
Point 10: 46-56: Fundamental problem is that surgical severity and inflammatory response that may adversely effect outcome is not always related to nociceptive input. Also, nociceptive input is not the major source of stress response and systemic inflammation that may cause detrimental outcome. Likewise, the link between nociception and inflammation and consequently complications is unclear. Therefore, the whole paragraph should be rephrased to indicate that despite these uncertainties, nociception monitors may allow a glimpse of the surgery related stress response apart from its original purpose of monitoring the balance between nociception and proper anesthetic management.
Response 10: Thank you for your appropriate comment. We added the following sentences at the last paragraph of Introduction section; “One may think, however, that both surgical severity and inflammatory response that may adversely affect postoperative outcome is not always related to nociception, and also that nociception is not the major source of surgical stress response and systemic inflammation that may cause detrimental outcome. By filling the gaps between nociception, inflammation, and consequently postoperative complications, this review provides a new concept that nociception monitor may allow a glimpse of the surgery related stress response apart from its original role.”
We also modified Figure 1 with additional descriptions of the balance between inflammation and anti-inflammation to show how we filled gaps between nociception, inflammation, and complications, and stated the balance at the 4th and 5th paragraphs in the section 5.
Point 11: 109: qNOX is unique in that it does not rely on a measure of autonomic activity. Therefore, the remark regarding the Table 1 can be misleading as it describes nociception monitors using autonomic responses. Clear distinctions and descriptions should be provided regarding the monitors that rely on autonomic responses or not.
Response 11: To avoid misunderstanding, we deleted “and the qNOX using EEG and electromyography” in the revised manuscript.
Point 12: Before 165 (Such monitoring~), add a sentence that clearly states the following contents: Although these monitors are more sensitive in detecting autonomic changes in nociceptive stimuli than traditionally used parameters, such as BP or HR, there is currently no convincing clinical evidence in favor of these monitors in that regard.
Point 13: 167: it should be noted that the potential to detect surgical stress response would only be confined to nociception monitors detecting autonomic responses, theoretically.
Responses 12 and 13: Thank you very much for your valuable comment. We rewrote the last paragraph in the section 4 as follows; “Monitoring nociception during surgery under general anesthesia has been proposed to avoid or manage sudden hemodynamic changes or unexpected body movements by monitoring the balance between nociception and anti-nociception. Although these monitors are more sensitive in detecting autonomic changes in nociceptive stimuli than traditionally used parameters, such as BP or HR, there is currently no convincing clinical evidence in favor of these monitors in that regard. Since these nociception monitors reflect autonomic responses elicited by both the balance between nociception and anti-nociception and the balance between inflammation and anti-inflammation, such monitoring is anticipated to have further roles to play in optimizing patient outcomes by monitoring other aspects of nociception (e.g., surgical stress responses) (Fig. 1)”
Point 14: 193: Although it is displayed in the Table, a brief description regarding the assessed variables of the SPI should be added.
Response 14: We changed “SPI values also reportedly” to “SPI values, which are calculated using heartbeat intervals and plethysmographic pulse wave amplitude, also reportedly”.
Point 15: 194: Likewise, descriptions regarding the assessed variables of ANI and PPI should be added.
Response 15: We changed “ANI and PPI monitors have also been” to “ANI and PPI monitors, of which sources are heart rate variability and pupillometry respectively, have also been” in the revised manuscript.
Point 16: 220: reduce à exert negative influence on
Response 16: We changed “reduce” to “exert negative influence on” in the revised manuscript.
Point 17: 246: enhanced à Enhanced
Response 17: We changed “enhanced” to “Enhanced”.
Point 18: 257-260: there is no clear link between ERAS (including prehabilitation) and the utility of intraoperative nociception monitors in that regard. The sentence needs to be rephrased that the adequacy of ERAS may be monitored through nociception monitors, which need to be validated.
Response 18: We replaced the original sentence, “Since the reduction of surgical stress responses utilizing ERAS and/or prehabilitation programs decreases the incidence of postoperative complications, nociception monitors represent candidate monitoring system for surgical stress responses during surgery as a part of perioperative management.” to the revised sentence, “The adequacy of ERAS and/or prehabilitation programs may be monitored through nociception monitors, which need to be validated.”.
Point 19: 288-295: average NR values showed correlation with the order of the surgical risks, which was already known (and classified) as low-, intermediate, and high-risk before surgery. What additional clinical value does the nociception monitor provide in that regard? Please, comment.
Response 19: Since these sentences may lead misunderstanding, we deleted them and rewrote as follows; “Results from nociception monitors combining these autonomic responses are thus also expected to be associated with postoperative complications. Higher values of NR averaged from the start to the end of surgery (mean NR), which are calculated using heart rate, systolic blood pressure, and peripheral perfusion index, were reportedly associated with a higher incidence of major postoperative complications, defined as a Clavien-Dindo Class ≥ IIIa, after gastrointestinal surgery (Table 1) [81].”.
Point 20: 308: conversely was used twice. Delete the second one.
Response 20: We deleted it.
Point 21: 308: brief description of the assessed variables by the NoL should be included
Response 21: We added “where NoL values are based on photoplethysmography, galvanic skin response, temperature, and an accelerometer” in the revised manuscript.
Point 22: 317: it should be clearly noted that this notion also needs to be validated
Response 22: We changed the last sentence in the section 8 to” the adequacy of nociception monitor-guided multimodal general anesthesia may be relevant for suppressing surgical stress responses, which need to be validated”.
Point 23: 326: hemodynamic data à how about monitoring autonomic responses in general?
Response 23: We changed “using hemodynamic data” to “monitoring autonomic responses”.
Point 24: 330: before the start of the sentence ‘Since few studies~’, add a sentence that clearly describes the pros and cons of the monitors: Although the monitors more accurately reflect nociception than the traditionally used hemodynamic parameters such as blood pressure or heart rate, there is no solid evidence regarding any clinically relevant influence of these devices on patient outcome.
Response 24: We added “Although the monitors more accurately reflect nociception than the traditionally used hemodynamic parameters such as blood pressure or heart rate, there is no solid evidence regarding any clinically relevant influence of these devices on patient outcome.” in the conclusion section.

Reviewer 2 Report
This is a very interesting topics that highlights a major medical problem. It's an evidence that the surgical aggression provokes a neuroendocrine and inflammatory reaction with a major impact in the postoperative rehabilitation. This review with the current data of the literature tries to highlight this major problem. The article is well structured.
This topics is becoming even more important nowadays with the concept of opioid low anaesthesia (OLA) and opioid free anaesthesia (OFA).
It is really important the scientific argumentation of the monitoring of nociception knowing that opioids used during operations spare the organism from the surgical aggression but their excessive consumption induces important undesirable effects.
Two small remarks for the authors:
- We can further enrich figure 1.
- You have done a very good literature synthesis of this problem. Can you try to give an opinion how we should orientate the scientific research in the future.
Author Response
Response to Reviewer 2 Comments
Point 1: We can further enrich figure 1.
Response 1: Thank you very much for your comment. We modified Figure 1 with additional descriptions of the balance between inflammation and anti-inflammation to show how we filled gaps between nociception, inflammation, and post operative complications in this review.
Point 2: You have done a very good literature synthesis of this problem. Can you try to give an opinion how we should orientate the scientific research in the future.
Response 2: We stated a future direction of clinical research regarding to nociception monitor in the section 8 as follows; “Since intraoperative multimodal approaches, including use of regional anesthesia with avoidance of long-acting opioids, intraoperative normothermia, maintaining fluid balance, and restrictive use of drains and tubes, have been utilized to reduce postoperative complications by suppressing surgical stress responses [86, 87], the adequacy of nociception monitor-guided multimodal general anesthesia may be relevant for suppressing surgical stress responses, which need to be validated.”

Round 2
Reviewer 1 Report
Thank you for your extensive revision.